# A Newly Developed Algorithm for Cloud Shadow Detection—TIP Method

**Viktoria Zekoll \* [ID], Raquel de los Reyes [ID] and Rudolf Richter [ID]**

DLR, German Aerospace Center, D-82234 Wessling, Germany; raquel.delosreyes@dlr.de (R.d.l.R.); rudolf.richter@dlr.de (R.R.)
\* Correspondence: viktoria.zekoll@dlr.de

**Abstract:** The masking of cloud shadows in optical satellite imagery is an important step in automated processing chains. A new method (the TIP method) for cloud shadow detection in multi-spectral satellite images is presented and compared to current methods. The TIP method is based on the evaluation of *thresholds*, *indices* and *projections*. Most state-of-the-art methods solemnly rely on one of these evaluation steps or on a complex working mechanism. Instead, the new method incorporates three basic evaluation steps into one algorithm for easy and accurate cloud shadow detection. Furthermore the performance of the masking algorithms provided by the software packages ATCOR ("Atmospheric Correction") and PACO ("Python-based Atmospheric Correction") is compared with that of the newly implemented TIP method on a set of 20 Sentinel-2 scenes distributed over the globe, covering a wide variety of environments and climates. The algorithms incorporated in each piece of masking software use the class of cloud shadows, but they employ different rules and class-specific thresholds. Classification results are compared to the assessment of an expert human interpreter. The class assignment of the human interpreter is considered as reference or "truth". The overall accuracies for the class cloud shadows of ATCOR and PACO (including TIP) for difference areas of the selected scenes are 70.4% and 76.6% respectively. The difference area encompasses the parts of the classification image where the classification maps disagree. User and producer accuracies for the class cloud shadow are strongly scene-dependent, typically varying between 45% and 95%. The experimental results show that the proposed TIP method based on thresholds, indices and projections can obtain improved cloud shadow detection performance.

**Keywords:** Sentinel-2; cloud shadow masking; TIP method; PACO; ATCOR

## 1. Introduction

The Earth has annual cloud coverage of approximately 70% [1]. Such high cloud coverage in multi-spectral satellite images, for example, those taken by Sentinel-2, can be seen as a negative effect on many remote sensing tasks [2]. This inevitable contamination limits the ability of a continuous observation of any location on the Earth and degrades the information that can be extracted from a scene [3]. Especially for land applications, the number of scenes with usable data is of high importance. For example, because the timing is important (crop yield estimation) or because the scene is not free-of-charge and one has to pay for the next acquisition if the current one contains a cloud shadow over the location of interest. A customer of such applications might be interested in determining the ground properties [4–7], or use the cloud and cloud shadow-free image for geological applications [8] and crop monitoring tasks [2]. Therefore, the correct and exact masking of clouds and cloud shadows is an important preprocessing step for atmospheric correction and is also required for shadow removal in multi-spectral imagery.

Several methods of shadow detection have been published over the past few years. Most of these methods are from the field of computer vision. In the work proposed by [9], the shadows appearing in an image were detected by using derivatives of the input image and color invariant images. In another study, [10] shadows were detected from

monochromatic natural images using shadow-variant and shadow-invariant cues from illumination, textural and odd-order derivative characteristics.

In remote sensing images, the detection of clouds and their cloud shadows is mostly evaluated with complex multistage processes [3,11–14]. However, some satellite sensors do not have enough spectral information to properly select appropriate thresholds for cloud shadow detection that will hold for a variety of images. For example, the GaoFen-1 and Proba-V images only have three bands in the visible band and one band in the near-infrared band available [3,15]. Another team [16] tried to overcome this lack of spectral information by introducing cloud and cloud shadow detection into deep convolutional neural networks (DCNNs). Furthermore, [17] proposed a method for cloud shadow detection relying on land cover data support using a shadow probability generation method. In [18], the detection of cloud and cloud shadows was done solely by evaluating spectral indices (CSD-SI algorithm).

In this study, the cloud shadow detection was performed on multispectral satellite images covering the spectrum from blue to short-wave infrared, 2.2 μm. The newly presented method can overcome the disadvantages of only being applicable to one sensor, as is the case for [19], since only specific spectral channels have to be present (VNIR and SWIR), and it does not have to undergo complex multistage processes as done by [3,11–14].

To understand the basic concept of shadow detection prior to atmospheric correction, it is important to know the basics of the radiation components and the relationship between the at-sensor radiance, also named top-of-atmosphere (TOA) radiance, and the digital number (DN). In the current state of the art PACO/ATCOR model, the input data are the recorded scene plus a meta file containing the acquisition date, time, solar and sensor view geometry, etc. The input data are given as scaled TOA radiance, named digital number (DN) [20].

For each spectral band of a sensor, a linear equation exists which describes the relationship between the DN (or brightness) and the TOA radiance, L. It is obtained with the radiometric offset, $c_0$, and gain, $c_1$, as follows, and has units of $W/(m^2 * sr * \mu m)$:

$$L = c_0 + c_1 DN \tag{1}$$

For some instruments, e.g., Sentinel-2, the data are given as top-of-atmosphere (TOA) reflectance $\rho_{TOA}$, also called apparent or at-sensor reflectance, $\rho^*$:

$$\rho_{TOA} = \rho^* = \frac{\pi L d^2}{E_s \cos \theta_s} \tag{2}$$

where $d$ is the Earth–Sun distance in astronomical units, $E_s$ is the extraterrestrial solar irradiance and $\theta_s$ is the solar zenith angle [21].

In this study, the PACO atmospheric processor was used. This is the python-based version of ATCOR. For PACO, the input data were in L1C radiances in units of $mW/(cm^2 * sr * \mu m)$. If the input scene is given in terms of TOA reflectance, it has to be converted into TOA radiance by solving Equation (2) for $L$.

The 13 spectral bands of a Sentinel-2 scene are composed of four bands at 10 m, six bands at 20 m and three bands at 60 m spatial resolution. PACO performs the atmospheric correction using the spectral information in all bands by resampling them to a 20 m cube, yielding an image cube of 13 bands with a size of 5490 × 5490 pixels. This so called "merged cube" is a Sentinel-2 TOA cube that is considered in the rest of this study.

## 2. Methods

### 2.1. Current PACO Shadow Masking

PACO performs pre-classification prior to atmospheric correction, which classifies the pixels of the input data into different classes. This is an important step, since the following atmospheric correction relies on a correct pre-classification. PACO uses the spectral channels in the visible, near infrared (NIR) and short-wave infrared (SWIR) bands. Additionally, empirical thresholds on the TOA reflectance are used to determine the selection of the

pre-classification classes. In PACO, the basic classes are land, water, snow/ice, cloud (non-cirrus), cirrus cloud, haze, shadow and topographic shadow [20].

The class of shadow is currently calculated using spectral and geometric criteria, whereas the topographic shadows are classified with the digital elevation model (DEM) and the solar geometry.

To date, the cloud shadow masking incorporated in PACO performs very well under good conditions, but fails for Sentinel-2 images that are located in desert regions and have features such as dunes and bright sand, and for scenes mainly covered by thick ice. In this paper, a new cloud shadow masking method for PACO is presented which will overcome these disadvantages. In a first section, the new TIP method is introduced (see Section 2.2). The results obtained from the new method are shown in Section 3. In the final part of the paper, the new method implemented in PACO is compared with the cloud shadow detection of ATCOR through the validation done by [22] (see Section 4).

*2.2. TIP Method*

This study describes the newly developed masking criteria that overcome the drawbacks of the current cloud shadow masking. The TIP method is named after its step by step computation of the cloud shadow map. These steps include the evaluations of *thresholds*, *indices* and *projections*. The exact order of the cloud shadow detection within the TIP method is listed below and is explained in detail in Sections 2.2.2–2.2.6.

- **T**hreshold selection—(T).
- Difference **i**ndex SWIR-NIR (DISN)—(I).
- Normalized difference water **i**ndex. (NDWI-green) for water correction—(I)
- Elimination of small isolated pixels or patches and smoothing of borders.
- Elimination of shadows not corresponding to the present cloud **p**rojection. (P)

2.2.1. TIP Input Image and Preparation for Masking

In this new approach, the RGB bands (RGB = 665,560,443 nm) are extracted from the full spectral TOA reflectance cube ($\rho^*$). Then, the RGB image is re-ordered into a BGR image (with B = 490 nm being the first band) and the channels are mapped in the range from 0 to 255 through linear stretching between the minimum and maximum range.

2.2.2. TIP Masking: Threshold Selection

The first step of the shadow masking is the calculation of the mean and the standard deviation of the BGR channels of the converted input image. These mean and standard deviations are calculated only including the classes of cloud shadow pixels and clear pixels. Hence, an accurate threshold selection for the cloud shadow pixels can be determined from the signal pixel distribution histogram, which only includes the distribution from clear and cloud shadow pixels. In order to perform this step as accurately as possible, only the signal arriving from cloud shadow pixels and clear pixels is evaluated; the background, cloud and topographic shadow pixels are included in a mask and excluded from the calculation.

The mean and standard deviation are calculated for each band (B,G and R) across all remaining pixels, and the threshold is calculated as in the LAB algorithm described by [23], but slightly adapted to the Sentinel-2 Multi Spectral Instrument (MSI), an instrument covering the spectral range from 0.44 µm to 2.2 µm. The threshold is calculated for a confidence interval of $\sim$1 $\sigma$; hence, 66.67% of the values lying within this interval. This corresponds to a fraction of $\frac{1}{3}$. The threshold per channel, i, can be written as follows:

$$T_i = mean_i - \text{standard deviation}\left(\frac{i}{3}\right) \tag{3}$$

where *i* stands for the blue (B), green (G) and red (R) bands.

Now the shadow pixel selection is performed for two conditions:

- If the sum of the mean values in the green and red channel is less or equal to 45, then all the pixels that lie between 0 and 100, and 100 and $T_{red}$ (for the B, G and R channels, respectively) are selected as shadow pixels, and others as non-shadow pixels.
- Else, all pixels that lie between 0 and $T_{blue}$, $T_{blue}$ and 255, and 255 and $T_{red}$ (for the B, G and R channels, respectively) are selected as shadow pixels and others as non-shadow pixels.

The threshold of 100 for the blue and green channels was selected, since the values in the shadow region are lower, as mentioned by [23]. To not lose any cloud shadows in scenes with a lot of water bodies, a threshold of 45 for the sum of the mean values in the green and red channels was determined.

After profound evaluations of Sentinel-2 scenes, it was clear that most cloud shadows lie below the threshold of 100, and bright pixels, such as cloud pixels, lie above the threshold of 100. The threshold selection alone does not give satisfying results. Dark vegetation and water pixels are typical examples of miss-classified pixels if no additional processing steps are included. Hence, in the next sections, further fine tuning steps are described to improve the shadow map.

### 2.2.3. TIP Masking: DISN for Dark Vegetation Correction

In order to avoid the miss-classification of dark vegetation pixels, a new spectral index is presented for the cloud shadow detection. This index is defined as the *difference deshadowing index SWIR-NIR* (DISN). The two bands were chosen because healthy vegetation shows high reflectance in the NIR spectrum and dark areas, which might be confounded by cloud shadows showing high reflectance in the SWIR spectrum [24].

The spectral index calculation for the shadow map was evaluated and created through trials with multiple scenes that will be discussed in Section 3. The best result was obtained when calculating the difference in the TOA reflectance, $\rho^*$, between a SWIR2 band at around 2200 nm and a NIR band at around 860 nm (see Equation (4)).

$$DISN = \rho^*_{SWIR2} - \rho^*_{NIR} \qquad (4)$$

Once the DISN has been computed for the original image, the peak of the histogram of the DISN is evaluated. With this peak, a threshold is defined as follows:

$$threshold_{DISN} = peak_{DISN} + |peak_{DISN} * 0.30| \qquad (5)$$

Now all shadow pixels defined in Section 2.2.2 that satisfy $DISN < threshold_{DISN}$ are omitted from the shadow mask and classified as non-shadow pixels. It can happen that there is more than one peak in the histogram of the DISN. If this case applies, then the lower peak is selected for computation.

### 2.2.4. TIP Masking: NDWI-Green for Water Correction

For the water pixel detection and elimination, the histogram of the green normalized difference water index ($NDWI_{green}$) is used. The NDWI allows one to differentiate water from dry land. Water bodies are known to have low reflectance and strong absorption in the NIR wavelength range. The green band has relatively high reflectance for water bodies compared to the reflectance from cloud shadows [25]. Hence, the NDWI uses the infra-red and green bands of remote sensing images [26]. The NDWI green is given as [27]:

$$NDWI_{green} = \frac{\rho^*_{green} - \rho^*_{NIR}}{\rho^*_{green} + \rho^*_{NIR}} \qquad (6)$$

For each scene, a pixel distribution of the $NDWI_{green}$ is made. All pixels of the scene are included in the $NDWI_{green}$ computation. The distribution of this index is expected to be divided into two populations with two peaks. The pixels with the values around the first peak (low NDWI) include the shadow pixels, and the pixels with a $NDWI_{green}$ value around the second peak (higher NDWI) are water pixels. Hence, this makes it possible to differentiate

between dark water pixels and shadow pixels. The value of the valley to the right of the main peak is detected, and all pixels with a $NDWI_{green}$ value above this valley are omitted from the cloud shadow pixels and assumed to be part of the water classification.

If more then two peaks are present, the differentiation between water and shadows is done using the valley to the right of the main peak.

Examples of the water pixel elimination can be found in Section 3.2.

### 2.2.5. TIP Masking: Small Isolated Pixels or Patches and Smoothing of Borders

This fine-tuning step of the shadow masking removes all dark fields, vegetation or dark pixels that are too small to be part of a cloud shadow.

This is done by using the current shadow map and splitting it into separate patches ("labels") [28]. The size of each patch is calculated, so that the number of shadow pixels contained in each patch is known. Only patches that contain an area larger than $100 \times 100$ m, corresponding to 25 pixels at 20 m of Sentinel-2, are selected as shadows, and the rest are excluded from the shadow map. Finally, the shadow map is smoothed for the final cloud shadow border pixel correction. The smoothing is performed by selecting 5 pixels into each direction around the cloud shadow pixels to be smoothed.

### 2.2.6. TIP Masking: Cloud Projection

In order to remove the rest of the shadows that are wrongly classified as cloud shadows, the cloud projection is evaluated for each classified cloud shadow pixel. If the projected cloud belongs to a cloud pixel or to a pixel outside of the scene, then the shadow is considered a cloud shadow. The rest of the shadow pixels are excluded from the cloud shadow mask.

As shown by [29], the orthographic positions of the cloud and the cloud shadow can be expressed through geometric equations which include the cloud height, H; the satellite view zenith and azimuth angles, $\theta_v$ and $\phi_v$, respectively; and the sun zenith and azimuth angles, $\theta_s$ and $\phi_s$, respectively.

Since the cloud projection step in the TIP method calculates the cloud projection pixel for each detected cloud shadow pixel, the equations given by [29] are rearranged for the corresponding cloud projection pixel on the scene (see Equation (7)).

$$
\begin{aligned}
x_{projection} &= X_{shadow} - H * (tan(\theta_v)sin(\phi_v) - tan(\theta_s)sin(\phi_s)) \\
y_{projection} &= Y_{shadow} - H * (tan(\theta_v)cos(\phi_v) - tan(\theta_s)cos(\phi_s))
\end{aligned}
\tag{7}
$$

As recently shown by [30], the direction of the cloud shadow can be given by the apparent solar azimuth, $\phi_a$.

To fully estimate the correct location of the cloud with respect to the cloud shadow, the distance, d, between the pixel of the cloud shadow and the cloud projection on the image plane has to be calculated. For the distance estimation, the sun and viewing angles and the cloud height are needed [30].

The apparent solar azimuth and the distance, d, are calculated for all currently estimated cloud shadow pixels, and the mean is evaluated. In order to get the distance for the x and y coordinates of the cloud shadow pixel, $d_x$ and $d_y$ are calculated as follows (Equation (8)):

$$
\begin{aligned}
d_x &= sin(\phi_a) * d_{mean} \\
d_y &= cos(\phi_a) * d_{mean}
\end{aligned}
\tag{8}
$$

For the final location of the cloud pixel with respect to the estimated cloud shadow, the distance is added to each cloud shadow pixel (see Equation (9)).

$$
\begin{aligned}
X_{final} &= x_{projection} + \frac{d_x}{H} \\
Y_{final} &= y_{projection} + \frac{d_y}{H}
\end{aligned}
\tag{9}
$$

The estimation of the correct cloud height, H, is done with an iterative approach, since the cloud altitude is unknown. The cloud projection is calculated over a range of cloud altitudes (from 0.5 to 10 km) and stops, when the projected cloud shadow pixels have their maximum matches with the cloud and/or cirrus mask given by PACO.

## 3. Results

### 3.1. Data and Material for Training Set

To test the new TIP cloud shadow masking on a set of data, six Sentinel-2 (S2) scenes were chosen. A list of the investigated Sentinel-2 A and B scenes is given in Table 1. The scenes were selected to cover different regions of Europe, Africa and Antarctica. Hence, the very different desert, ice and continental climates were investigated. They represent flat and mountainous sites with cloud cover from 3% to 80% and include the presence of cumulus, thin and thick cirrus clouds. Additionally, the scenes represent different land cover types, such as desert, urban, cropland, grass, forest, wetlands, sand and coastal areas. The range of solar zenith angles is from 27° to 67°. Only the results of 3 scenes (Gobabeb, Arcachon, France, Barrax-2) are shown in Section 3.2 as examples.

**Table 1.** Sentinel-2 level L1C test scenes. Information on scene climate, main surface cover, rural/urban. (SZA = solar zenith angle).

| Scene | Location | Date | Tile | SZA | Desert | Ice/Snow | Nonveg | Veg | Water | Mountains | Rural | Urban |
|---|---|---|---|---|---|---|---|---|---|---|---|---|
| 1 | Africa, Gobabeb | 2019/03/06 | T33KWP | 32.5° | X | | | | | X | X | |
| 2 | Antarctic | 2018/01/26 | T34DFH | 58.8° | | X | X | | X | | | |
| 3 | France, Arcachon | 2017/11/15 | T30TXQ | 63.9° | | | X | X | X | | X | X |
| 4 | France | 2016/01/16 | T31TFJ | 66.8° | | | X | X | X | X | X | X |
| 5 | Morocco, Quarzazate | 2018/08/30 | T29RPQ | 27.2° | X | | X | X | X | | | |
| 6 | Netherlands, Amsterdam | 2018/09/13 | T31UFU | 49.7° | | | | X | X | | X | X |

### 3.2. Masking Sequence of TIP Method

This section presents the results of the TIP method explained in Section 2.2. Figure 1 shows the results of the cloud shadow masking of the TIP method on the Gobabeb scene. The left image shows the cloud shadow mask provided by the current ATCOR masking algorithm. The middle image shows the cloud shadow mask obtained through the TIP method. The image on the right of Figure 1 shows the original Gobabeb scene, which was stretched into the RGB = 665/560/490 nm bands for better optical comparison. As can be seen in Figure 1, the TIP method did improve the cloud shadow masking in the desert scene. The cloud shadows present in the right lower corner were not detected by ATCOR but were detected with the TIP method. Furthermore, ATCOR faced some difficulties in the top part of the cloud shadow mask. This can be seen in form of stripes and led to multiple misclassified pixels.

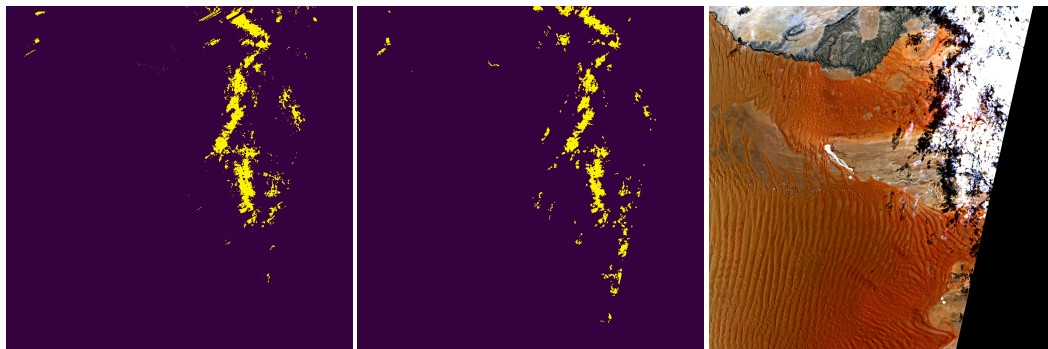

**Figure 1.** Gobabeb shadow map. Left: cloud shadow map of ATCOR; middle: TIP cloud shadow map; right: Gobabeb RGB = 665/560/490 nm with linear histogram stretching.

To demonstrate the importance of the TIP method masking step that corrects for misclassified water pixels using the NDWI-green threshold, as mentioned in Section 2.2.4,

a Sentinel-2 scene from Arcachon was used. This scene acquired on 15 November 2017, is of a place located on the seaside that has a lot of water bodies (see Figure 2).

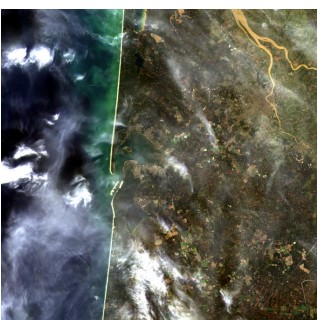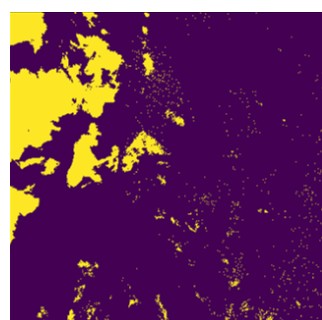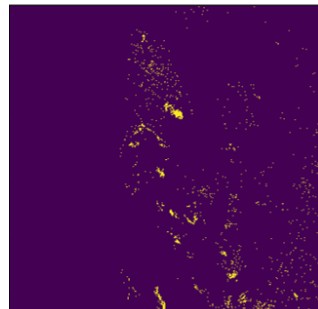

**Figure 2.** Arcachon scene analysis of the NDWI. The left image represents the original TOA radiance image. The figure in the middle represents the shadow mask before the water correction. The figure on the right represents the shadow mask after the water correction.

The left image of Figure 2 shows the true color image of the scene. The middle image represents the shadow mask before the water correction and the right image the shadow mask after the water correction. All major water bodies seen in yellow in the middle image of Figure 2, not excluded previously, were excluded from the shadow map.

## 4. Validation of Results

### 4.1. Validation Statistic

In order to quantitatively validate the results obtained from the TIP method, the validation done by [22] was used, where twenty Sentinel-2 scenes were processed with three different mono-temporal masking codes. These masking codes are *Function of mask* (Fmask FORCE version 3.0), ATCOR (version 9.3.0) and Sen2Cor (version 2.8.0). The scenes were selected to cover all continents in order to represent different climates, seasons and weather conditions. This enabled us to cover a wide range of land cover types (desert, urban, cropland, grass, forest, wetlands, sand, coastal areas, glaciers). The range of solar zenith angles was from 18° to 62°. No solar zenith angle above 70° was chosen, since ATCOR and Sen2Cor have this angle as their upper limits. This is due to extreme BRDF effects for angles above this limit, and the scenes are mostly covered by shadows from buildings, trees, etc. Furthermore, the validation aimed for an amount of 1000 randomly selected samples per image with a minimum of 50 samples for the smallest class [31–34].

The validation in this section compares the performance obtained by the ATCOR algorithm in [22] with the newest masking code of PACO (revision 1432), which includes the TIP method for the cloud shadow class.

The reference data were slightly changed, since more pixels were validated than in the comparison performed by [22]. Therefore, the exact same validation steps were performed, and the statistical test was repeated for the ATCOR masking algorithm. The results consist of a confusion matrix with the number of correctly classified pixels in the validation set. This confusion matrix enables the computation of the *user accuracy* (UA), *producer accuracy* (PA) and *overall accuracy* (OA) of classification.

#### User, Producer and Overall Accuracy

The UA for a specific class gives the percentage of the area mapped as this class that has reference of the class. The PA of a specific class gives the percentage of the area of the reference class that is mapped as this class. The OA of a specific class is the total number of correctly classified pixels of this class divided by the total number of reference pixels of this same class [35,36]. Hence, the UA and PA are the important accuracies for fully understanding the statistics. The OA will not add any new statistical information if UA and PA are known. Hence, one can claim that the OA will partially hide the lack of accuracy present in the UA or PA.

### 4.2. Sentinel-2 Validation Results

Tables 2 and 3 provide the results for absolute validation of the classifications of the ATCOR and PACO masking algorithms, which are comparable to the results presented in [22].

**Table 2.** Summary of classification accuracy (percent) for difference area (A = ATCOR, P = PACO; bold face numbers indicate the best performances).

| Class | UA (A) | UA (P) | PA (A) | PA (P) |
|---|---|---|---|---|
| clear | 74.6 | **75.7** | 67.1 | **80.4** |
| semi-transp. cloud | **61.1** | 53.8 | 35.8 | **38.8** |
| cloud | 62.2 | **80.5** | **67.4** | 47.6 |
| cloud shadow | **69.0** | 57.3 | 64.9 | **75.3** |
| water | 48.2 | **91.5** | **82.3** | 81.3 |
| snow/ice | **60.5** | 51.4 | **67.2** | 66.9 |
| topographic shadows | **32.0** | 20.6 | 1.2 | **2.4** |

**Table 3.** Summary of the cloud shadow class's overall accuracy (percent) (A = ATCOR, P = PACO; bold face numbers indicate the best performances).

| | | OA Difference Area | |
|---|---|---|---|
| **Scene** | **Location** | **A** | **P** |
| ID | Average (all scenes) | 70.4 | **76.6** |
| 1 | Antarctic | 0 | 0 |
| 6 | Estonia, Tallin | **85.0** | 84.5 |
| 8 | Italy, Etna | 63.5 | **83.5** |
| 9 | Kazakhstan, Balkhash | 57.5 | **78.5** |
| 10 | Mexico, Cancun | **64.5** | 42.5 |
| 11 | Morocco, Quarzazate | 91.5 | **94.5** |
| 12 | Mosambique, Maputo | 7.5 | **46.5** |
| 13 | Netherlands, Amsterdam | 65.0 | **76.0** |
| 14 | Phillipines, Manila | 66.0 | **71.0** |
| 16 | Russia, Yakutsk | 73.0 | **88.0** |
| 17 | Spain, Barrax-1 | 67.0 | **78.5** |
| 18 | Spain, Barrax-2 | 76.0 | **87.5** |
| 19 | Switzerland, Davos | 52.5 | **55.5** |
| 20 | USA, Rimrock | 48.5 | **67.5** |
| 2 | Argentina, Buenos Aires | **99.8** | 93.7 |
| 3 | Australia, Lake Lefroy | **100.0** | **100.0** |
| 4 | Bolivia, Puerto Siles | **99.8** | 98.4 |
| 5 | China, Dunhuang | 92.3 | **92.4** |
| 7 | Germany, Berlin | **99.7** | 94.0 |
| 15 | Russia, Sachalin | **99.2** | **99.2** |

A masking algorithm which will satisfy not only the producer but also the user aims to obtain high values in both PA and UA. Such an algorithm produces a classification map that detects as many pixels of each class as possible, but misclassifies as few pixels as possible per class.

Table 2 provides a summary of the UA and PA per class, i.e., the averages over all 20 scenes. If we concentrate our error matrix in Table 2 on the classification of cloud shadows, we can draw following conclusions for the classification done by PACO: The cloud shadow classification is 75.3% accurate. This high value could lead a producer to conclude that this classified Sentinel-2 image is sufficiently accurate for their needs. However, a user will be disappointed to find that only 57.3% of the pixels identified as cloud shadows on the classification map are actually cloud shadow pixels. Hence, 75.3% of the cloud shadows have been correctly identified as such, but only 57.3% of those

areas identified as cloud shadows are actually cloud shadows, while 42.7% of those areas identified as cloud shadows belong to another class.

It can be noted that Table 3 is divided into three sections. The top section contains the OA of the class cloud shadow of all the scenes for each masking algorithm. The middle section contains the OA of the class cloud shadow per scene for each masking algorithm and for the scenes where cloud shadows are present. The bottom section of Table 3 shows the false positive percentages for the scenes that do not contain any cloud shadows. For this type of scene, the reference statistics have been changed, and since no cloud pixels were expected, the accuracy of the false positives was evaluated as such: 100.0% means that no pixels were wrongly classified as cloud shadows. Depending on the misclassified pixels, the OA will deviate from 100.0%. Boldface numbers indicate the method with the best performance. Overall, PACO have the highest OA over all scenes, with 76.6%. Following PACO is ATCOR, with an OA of 70.4%. Hence, PACO has improved the ATCOR results for cloud shadows.

Only the scenes of Mexico and Antarctica caused worse results for PACO compared to ATCOR. This was due to the lack of correct cloud and water maps and will be discussed in Section 5.

Table 4 shows the UA and PA of the cloud shadows for each scene, for ATCOR (A) and PACO (P). The Table shows that PACO performs best in PA for the class cloud shadow and has a high OA, since the values for the UA are better than those of ATCOR in many cases. ATCOR has a high UA for the class cloud shadow, but a lower PA than PACO in all scenes apart from Mexico. The right column of Table 4 furthermore indicates the amounts of cloud shadow pixels that have been annotated in the reference data. The scenes that do not contain any cloud shadows are listed in a separate section within Table 4. In this section, the UA and PA for the false positive pixels of the class cloud shadows are shown. It can be seen only in the UA of all masking algorithms when pixels were wrongly classified as cloud shadows, and hence the accuracy deviates from 100.0%.

**Table 4.** Summary of cloud shadow class user and producer accuracies (percent) (A = ATCOR, P = PACO; bold face numbers indicate the best performances).

| Scene | Location | UA (A) | UA (P) | PA (A) | PA (P) | Annotated Cloud Shadow Pixels |
|---|---|---|---|---|---|---|
| 1 | Antarctic | 0 | 0 | 0 | 0 | 35 |
| 6 | Estonia, Tallin | **93.0** | 87.0 | 77.0 | **82.0** | 1415 |
| 8 | Italy, Etna | **88.0** | 80.0 | 39.0 | **87.0** | 1446 |
| 9 | Kazakhstan, Balkhash | 76.0 | **93.0** | 39.0 | **64.0** | 809 |
| 10 | Mexico, Cancun | **91.0** | 60.0 | **38.0** | 25.0 | 755 |
| 11 | Morocco, Quarzazate | 88.0 | **93.0** | 95.0 | **96.0** | 32,318 |
| 12 | Mosambique, Maputo | **4.0** | 3.0 | 11.0 | **90.0** | 82 |
| 13 | Netherlands, Amsterdam | **77.0** | 68.0 | 53.0 | **84.0** | 857 |
| 14 | Phillipines, Manila | **74.0** | 58.0 | 58.0 | **84.0** | 1076 |
| 16 | Russia, Yakutsk | 80.0 | **96.0** | 66.0 | **80.0** | 958 |
| 17 | Spain, Barrax-1 | 87.0 | **93.0** | 47.0 | **64.0** | 1592 |
| 18 | Spain, Barrax-2 | 73.0 | **86.0** | 79.0 | **89.0** | 25,115 |
| 19 | Switzerland, Davos | 61.0 | **62.0** | 44.0 | **49.0** | 2561 |
| 20 | USA, Rimrock | 45.0 | **69.0** | 52.0 | **66.0** | 847 |
| 2 | Argentina, Buenos Aires | **99.6** | 87.4 | **100.0** | **100.0** | 0 |
| 3 | Australia, Lake Lefroy | **100.0** | **100.0** | **100.0** | **100.0** | 0 |
| 4 | Bolivia, Puerto Siles | **99.5** | 96.8 | **100.0** | **100.0** | 0 |
| 5 | China, Dunhuang | 84.6 | **84.8** | **100.0** | **100.0** | 0 |
| 7 | Germany, Berlin | **99.3** | 88.1 | **100.0** | **100.0** | 0 |
| 15 | Russia, Sachalin | 98.3 | **98.5** | **100.0** | **100.0** | 0 |

To further compare the classification performances of ATCOR and PACO, the classification maps for a specific scene and a subset taken from that scene are shown in

Figures 3 and 4. Figure 3 shows scene number 18 from Spain (Barrax) taken on 19 March 2017 with a zenith angle of 22.0° and a azimuth angle of 143.2°. In Figure 4, a subset of scene ID 18 can be found. It nicely illustrates the improvements that the TIP method adds to PACO with respect to the current ATCOR version. It also demonstrates where each of the different algorithms puts its focus. While ATCOR shows a very high PA for the cloud mask, PACO has the highest PA for the cloud shadows. This is due to high importance of the TIP method reaching a high PA for the class cloud shadow for further cloud shadow removal. ATCOR, on the other hand, focuses more on a high UA for cloud shadows, and hence does not classify cloud shadows very carefully at the cloud shadows' borders. PACO is designed to classify all the border pixels as cloud shadows.

The confusion within and between classes can be illustrated in spider diagrams using the proportions of the individual class omissions for the difference area (Figure 5). The difference area encompasses the parts of the classification images where the classification maps provided by ATCOR and PACO disagree. Validation statistics over the difference area gives a good comparison between the strengths and weaknesses of each processor [22].

Figure 5 illustrates spider diagrams for the omission and commission for difference areas of the classes clear and cloud shadows.

From the left upper image of Figure 5, it can be noted that ATCOR had the most omissions of clear pixels as water pixels. PACO caused omissions of clear pixels as cloud shadows. This is due to the higher PA of PACO for cloud shadows. Due to PACO wanting to classify as many cloud shadow pixels from the shadow border as possible, the PA is higher than its UA. This can also be seen in the lower left image of Figure 5, where PACO has ommissions of clear pixels as cloud shadows.

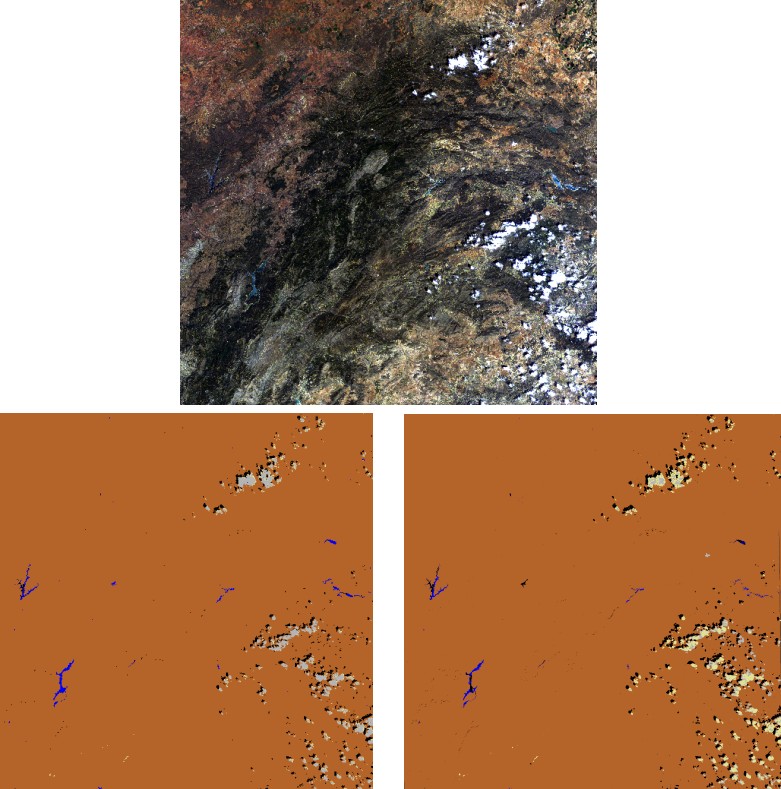

**Figure 3.** Top row: true color (RGB = 665, 560, 443nm) composite of scene ID 18 (Barrax-2). Bottom row (**left** to **right**): ATCOR and PACO classification maps (with clear (brown), cloud (grey), water (blue), shadow (black) and cirrus cloud (yellow) pixels).

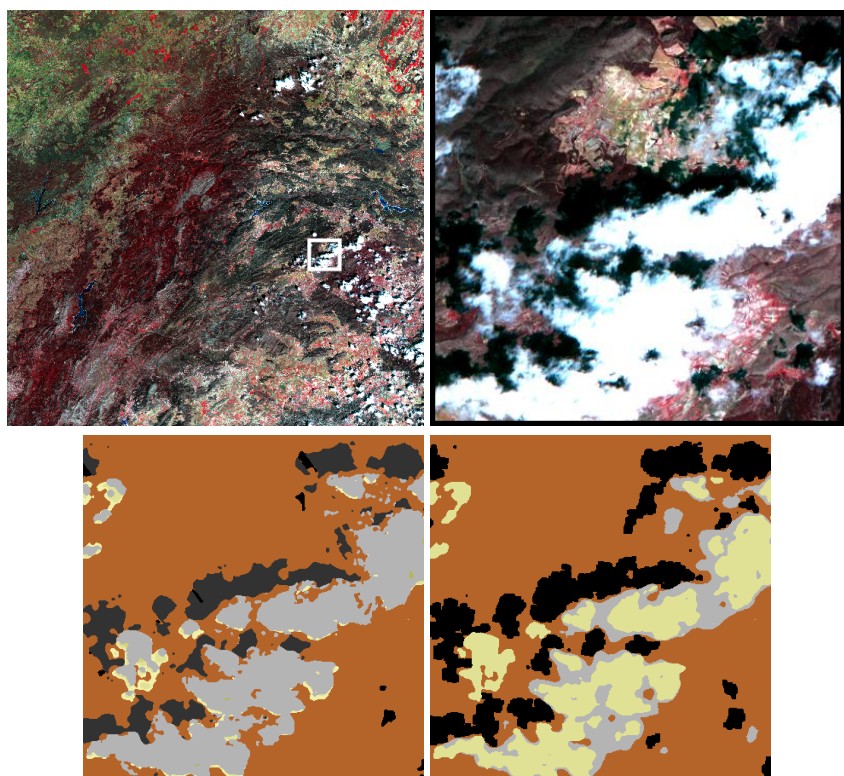

**Figure 4.** Top row (**left** to **right**): CIR (RGB = 865, 665, 560nm) composite and CIR subset of scene ID 18 (Barrax-2). Bottom row (**left** to **right**): ATCOR and PACO classification maps of the subset (with clear (brown), cloud (grey), shadow (black) and cirrus cloud (yellow)).

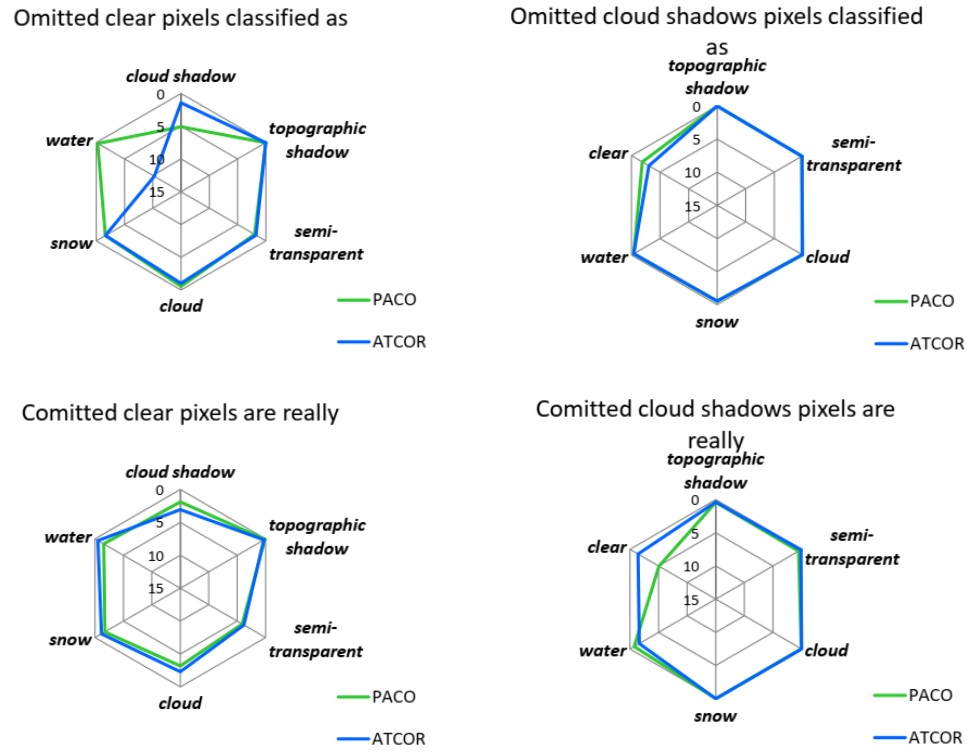

**Figure 5.** Omission and commission per Class for difference areas for the classes clear and cloud shadows. PACO is represented in the color green and ATCOR in the color blue.

Furthermore, the differences in the UA and PA between ATCOR and PACO can be seen in the two images on the right of Figure 5. The top right image shows that ATCOR had more omissions of cloud shadow pixels as clear pixels. This corresponds to its higher UA. The bottom right image shows that PACO had more ommissions of cloud shadow pixels as clear pixels, which on the other hand, corresponds to its higher PA. PACO has, therefore, a less conservative cloud shadow mask than ATCOR. This is due to PACO wanting to reach high values in PA in order to proceed with the shadow removal process. ATCOR, as proven by the omission of cloud shadow pixels, has a more conservative cloud shadow map. Hence, it has a lower PA value and will not be able to detect all cloud shadow pixels around the border of a shadow.

## 5. Discussion

Remotely sensed optical images have many details to be accounted for and need a lot of fine-tuning than in computer vision applications. The LAB method alone, as done by [23], might be a good method for simple close-range images taken by an RGB camera. However, it is not possible to adapt the LAB color scale to detect shadows in a Sentinel-2 satellite image. Nevertheless, the threshold selections implemented in the TIP method have opened new doors for the shadow detection. The presented TIP method adapts thresholds based on histogram analysis and introduces new spectral criteria to improve the shadow mask. Furthermore, it incorporates projection methods for the cloud shadows, which has so far only been used for the detection of clouds.

To compare how the cloud shadow detection of the TIP method incorporated into PACO differentiates from other masking algorithms, the analysis done by [22] was performed once again to compare the PACO masking algorithm to ATCOR. This validation done by the human eye has to be seen as a guidance for a solid comparison between different algorithms. Furthermore, each algorithm has a different focus on user or producer accuracy. The cloud shadow mask from the TIP method included in PACO is designed to detect as many cloud shadows as is realistic in order to perform a subsequent cloud shadow correction. This is also the reason why the PA (75.3%) is much higher than the UA (57.3%) (see Table 2). A lower UA has as a consequence that clear pixels close to the border of the cloud shadows will be deshadowed in shadow removal algorithms following the shadow masking. On the other hand, Sen2Cor, as evaluated by [22], had the highest UA for the class cloud shadow of 81.0%, since it is set to obtain as few false positive pixels as possible, and therefore has a more conservative cloud shadow masking. However, this happens at the expense of the PA percentage, which is lower than the PA of PACO.

Another criterion which differs between masking algorithms is when multiple classes lie on top of each other. For example, if cloud shadows are over water, then the pixels are identified as water for PACO. This is because the priority in PACO is that water bodies are correctly identified, and they are not excluded if parts are covered by shadows. Hence, the TIP method excludes all water pixels before starting the cloud shadow calculation. This has also the consequence that if a water body is wrongly classified in the steps prior to the TIP masking, then cloud shadows might be wrongly excluded from the final cloud shadow mask. This was the case of scene number 10 from Mexico imaged on 27 May 2018. Not all cloud shadows have been detected by the TIP method due to wrong cloud and water classification.

Another critical scenario for all cloud shadow masking algorithms is when the sensor incidence angle is very low. The scene from Antarctica is such a case. Here, the incidence angle of the sensor is 4.9°, close to nadir. Due to the low incidence angle, the scene appears very dark in regions where no snow and clouds are present. However, since the scene has a lot of ice and clouds, these areas appear very bright with respect to areas where no ice or clouds are present. PACO has difficulty determining the clouds that lie above the ice. Hence, due to the cloud projection step present in the TIP method, the shadows from the clouds above the ice are removed from the cloud shadow map. Correct classification of clouds is important for the cloud shadow detection within the TIP method. A wrong cloud mask will lead to errors in the cloud shadow detection.

Furthermore, one has to note that the TIP method can be used for VNIR-SWIR sensors, such as EnMAP [37], PRISMA [38], Landsat-8 [39] and Landsat-9 [40]; but for VNIR sensors, such as the DESIS sensor [41], the method can only be implemented partially. In the case of DESIS, the DISN correction explained in Section 2.2.3 will have to be excluded.

The newly presented TIP method overcomes the disadvantage of only being applicable to one sensor, as is the case for Fmask4P, which is only applicable for Sentinel-2 [19]. Fmask5, on the other hand, can be applied to Landsats 4–8 and Sentinel-2 [42]. The newly presented method only needs the presence of specific spectral channels (VNIR and SWIR) and can be applied to hyperspectral sensors such as PRISMA [38] and EnMAP [37]. Furthermore, the TIP method does not have to undergo complex multistage processes, as is currently done in most state-of-the-art methods [3,11–14]. Additionally, the method combines the advantages of multiple spectral indices that can be used for the detection of cloud shadows.

Even though the TIP method has only been tested for Sentinel-2 scenes with a geometric resolution of 20 m, it can be assumed to work for Landsat-8, which has a resolution of 30 m, since Fmask5 [42] works without any issues for Landsat-8 and Sentinel-2. One has to note that for sensors with ground sampling distances of 1–5 m, additional problems will arise even when all VNIR and SWIR channels are present. Hence, more research will have to be done in the future in order to be able to include such sensors.

## 6. Conclusions

A newly developed cloud shadow masking method called TIP, which has been tested in the PACO masking algorithm, was proposed. Cloud shadows of multispectral satellite images covering the spectrum 0.44 μm to 2.2 μm are found through the steps, which include the evaluation of thresholds, indices and projections. This method is intended to improve the PA of the class cloud shadow of the current ATCOR and PACO masking algorithms, because it is intended to be a prior step to a deshadowing method. Furthermore, the performance for the class cloud shadows of the improved PACO version was statistically compared with that of ATCOR for S2 scenes. The overall accuracies for the class cloud shadow were 70.4% and 76.6% for ATCOR and PACO, respectively. Hence, the newly developed TIP method for cloud shadows has improved the current cloud shadow mask. The TIP method encounters limitations for scenes where the cloud and water maps are wrongly classified prior to the cloud shadow detection. Future improvements could involve the fine-tuning of the cloud and water masks, which will improve the results of the TIP method for cloud shadows. Additionally, the TIP method cannot be fully implemented for VNIR sensors such as DESIS [41], but is applicable for VNIR-SWIR sensors such as EnMAP [37], PRISMA [38] and Landsat-8 [39], Landsat-9 [40].

**Author Contributions:** Conceptualization, V.Z.; Data curation, V.Z.; Formal analysis, V.Z.; Investigation, V.Z.; Methodology, V.Z.; Resources, R.d.l.R. and R.R.; Software, V.Z.; Supervision, R.d.l.R. and R.R.; Validation, V.Z.; Writing—original draft, V.Z.; Writing—review & editing, R.d.l.R. and R.R. All authors have read and agreed to the published version of the manuscript.

**Funding:** This research received no other external funding.

**Conflicts of Interest:** The authors declare no conflict of interest.

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
