# Peer review of "A Newly Developed Algorithm for Cloud Shadow Detection—TIP Method"

_remotesensing, doi:10.3390/rs14122922_

Round 1
Reviewer 1 Report
In section 2.2 (lines 107-108) it is not clear how the mean and standard deviation is computed. I assume that a scene classification (cloud masking) is performed before this step. Could you please clarify?
In table 1, not sure you have also considered cases where cloud shadows are completely detached from the originating clouds. In addition, also cases with SZA > 70â—¦. Could you please confirm or clarify?
In Figure 1, seems that in the upper-centre part of the image classified by TIP a wrong cloud shadow is detected (it is not present on ATCOR classification). Could you please explain the reason of this false positive?
In Figure 2, seems there is an error in the label, right image is not an NDWI histogram but the shadowed mask after water correction. Please correct it.
Please add the colours legend for the classification map on Figures 3 and 4.
Author Response
Dear reviewer,
thank you very much for your comments on my paper. I have tried to adapt the paper to your comments and hope that it is satisfactory for you.
To reply to each of your comments:
- I have clarified lines 107-108 in section 2.2
- I have also added an explanation why only angles below 70 degrees are used in section 4.1
- thank you very much for pointing out the wrong label in figure 2. I have corrected the label.
- I have added a colour legend for figure 3 and 4 into the label.
- in figure 1, you are correct that there are still false positives in the TIP method. This is the future work that will have to be done to improve the last false positive present. The cloud projection is one of the areas that will be revisited in order to be as accurate as possible. Since if false positives lie close to a cloud, the are misclassified as being part of this cloud.
Please let me know if you have any more comments.
Kind regards,
Viktoria Zekoll
Reviewer 2 Report
Authors present a new method (TIP method) for cloud shadow detection in multispectral satellite images.
The study is very important in the field of remote sensing but some changes must be taken into account before it is considered for publication:
- The abstract should highlight the importance of the methodology with respect to other existing and already contracted methodologies.
- The introduction is poor. Recent studies on cloud detection methodologies should be included, explaining in detail their advantages and then contrasting them with those presented in the study.
- It would be interesting to include a map with the areas used for the study.
- Statistically justify the sample size used in the study. Why is this number of images used?.
- The discussion section is poor and should be expanded, highlighting in detail the main advantages of the methodology used compared to others already used by other authors. This section is crucial to give importance to the study.
Author Response
Dear reviewer,
thank you very much for your comments on my paper. I have tried to adapt the paper to your comments and hope that it is satisfactory for you.
To reply to each of your comments:
- I have added the highlighted sections to the abstract in order to highlight more the importance of my method.
- for the introduction I have added more recent studies and also added some explanation about the advantages compared to the state-of-the-art methods.
- I have not added a map of the areas that have been studied. This decision was taken, since I have compared my method exactly with the method done in the referenced paper. This paper shows the areas that have been studied. Hence, due to paper number limitation the reader is welcome to have a look at the map in the referenced publication. I hope that this is a decent argument for you.
- To the statistical reasoning of the number of images used, an abstract was added. Furthermore the TIP method is compared exactly after the paper referenced, so this means that exactly the same reference scenes are taken for validation in order to be as consistent and exact as possible.
- I have added more details to the discussion in order to clarify the advantages and disadvantages of my presented method. I hope that this is more satisfactory for you now.
Please let me know if you have any more comments.
Kind regards,
Viktoria Zekoll
Round 2
Reviewer 2 Report
All the suggestions are incorporated by authors.